# Renin-angiotensin-aldosterone system variations in type 2 diabetes mellitus patients with different complications and treatments: Implications for glucose metabolism

**Ningning Wang[1], Junhui Li[1], Erjun Tian[1], Shutong Li[1], Shuai Liu[2,3,4], Fei Cao[5], Junfeng Kong[1]\*, Baohong Yue [2,3,4]\***

**1** Department of Laboratory Medicine, The First People's Hospital of Pingdingshan, Pingdingshan, Henan, China, **2** Department of Laboratory Medicine, The First Affiliated Hospital of Zhengzhou University, Zhengzhou, Henan, China, **3** Faculty of Laboratory Medicine, The First Clinical Medical College, Zhengzhou University, Zhengzhou, Henan, China, **4** Key Clinical Laboratory Medicine of Henan Province, Zhengzhou, Henan, China, **5** Department of Orthopedics, The First People's Hospital of Pingdingshan, Pingdingshan, Henan, China

\* fccybh@zzu.edu.cn (BY); 13393796530@126.com (JK)

## Abstract

### Background

The presence of hypertension and various acute or chronic complications may affect the renin-angiotensin-aldosterone system (RAAS) in patients with type 2 diabetes mellitus (T2DM), which plays a crucial role in the regulation of glucose metabolism. However, the quantitative distribution of the RAAS components in relation to the progression of T2DM and the treatment of hyperglycemia and hypertension, as well as their association with different stages of complications and glucose metabolism, has not been well studied.

### Methods

We enrolled a total of 151 patients with T2DM and essential hypertension, 40 patients with T2DM and normotension, and 46 healthy controls in the study. They were categorized into subgroups based on criteria for diabetic complications. Statistical analyses, including Spearman rank correlation and multiple linear regression, were conducted to assess the relationship between RAAS components and glucose metabolism indexes such as HbA1c, FBG, CP, HOMA-β, HOMA-IR, and UACR.

### Results

The results revealed significant differences in AII, ALD, REN, and ARR levels across various complication subgroups. Notably, the concentrations of ALD and REN exhibited a consistent trend, while ARR showed an opposite trend to the REN concentration. More than 60% of hypertensive patients were treated with ACEI/ARBs and calcium channel blockers, while 29.8% of the patients were prescribed β-blockers, resulting in decreased

**Data availability statement:** All the data necessary to replicate this paper's findings can be obtained through the following public repository https://github.com/ningningwang688/RAAS-and-T2DM.

**Funding:** This study was supported by the National Natural Science Foundation of China (grant numbers 82074268 and 82474325). Author FC is a participant in these grant programs and contributed to project administration and manuscript review. The other authors have no relevant competing interests.

**Competing interests:** The authors have declared that no competing interests exist.

REN and increased ARR levels. All T2DM patients received antidiabetic treatment, among which 95 (49.7%) took SGLT-2is, 40 (20.9%) took GLP-1RAs injection and 55(28.8%) took DPP-4is. The subsequent analysis revealed that SGLT-2is, GLP-1RAs, DPP-4is and other glucose-lowering agents had no statistically significant effect on the RAAS system ($p > 0.05$). The correlation matrix analysis indicated positive associations between ALD, REN, CP, and HOMA-IR. Furthermore, the REN levels were negatively correlated with UACR in the hypertensive group and positively correlated with HbA1c and FBG levels in the normotensive group. Multiple linear regression analysis demonstrated that ALD levels increased with higher levels of CP and HOMA-IR, independently of the RAAS system, anti-RAAS treatment and antidiabetic therapy. REN levels decreased with increasing UACR and β-blocker usage in the hypertensive group, while they increased with higher levels of HbA1c, FBG, and HOMA-IR in the normotensive group, independently of the RAAS system and antidiabetic therapy.

## Conclusions

The activation status of the RAAS system varied among T2DM patients with different complications, highlighting the need for clinical differentiation. ALD was positively associated with insulin resistance and glucose metabolism impairment, while REN exhibited negative correlations with urinary microalbumin and β-blocker usage, and positive correlations with hyperglycemia and insulin resistance. Blocking the RAAS system holds promise for improving insulin sensitivity and β-cell function, and potentially reversing abnormal glucose tolerance or ameliorating glucose metabolism disorders.

## 1. Introduction

The worldwide population of individuals diagnosed with diabetes mellitus (DM) has surged to 537 million in 2021 [1] and is projected to escalate to 642 million by 2040 [2]. In China, the overall prevalence of diabetes has been on the rise, increasing from 9.7% in 2007 and 2010 to 10.4% in 2013, and further to 11.2% in 2017 [3–7]. China ranks number one with an estimate of 129.8 million adults with diabetes by 2017 [7]; among them, type 2 diabetes mellitus (T2DM) accounts for the vast majority (>90%) [8]. T2DM represents a group of metabolic disorders characterized by hyperglycemia, primarily caused by inadequate insulin secretion or resistance to insulin [9,10]. T2DM patients are often accompanied by hypertension, with rates ranging from 40% to 60% [8]. They frequently experience acute complications such as ketoacidosis, as well as various microvascular issues including retinopathy, nephropathy, and neuropathy [11]. Research indicates that the activation status of the renin-angiotensin-aldosterone system (RAAS) undergoes alterations in Type 1 and Type 2 diabetes mellitus (T1DM/T2DM) patients with various complications [12–14]. RAAS, a hormonal system that regulates blood pressure, is present in the kidneys, adrenal glands, blood vessels, nervous system, and pancreatic β-cell [12,15]. Studies have demonstrated that aldosterone (ALD) can impede insulin secretion and sensitivity, which are pivotal elements in the progression of T2DM [9,14]. Previous investigations have highlighted a higher prevalence of diabetes among patients with primary aldosteronism compared to those with essential hypertension [16]. Inhibiting the RAAS signaling pathway not only reduces blood pressure and insulin resistance but also provides renal protection in diabetic individuals [17]. Current data shows a continuous rise in new cases of diabetic kidney dialysis despite the increasing usage of RAAS blockers [18]. This trend may

be attributed to sub-optimal dose of RAAS blockers by patients or inadequate comprehensive assessment by doctors regarding the course of T2DM and the activation status of RAAS. Hence, precise detection of RAAS is crucial for accurately guiding treatment decisions.

A study by Griffin TP et al. indicated a positive correlation between increased renin (REN) levels and elevated HbA1c levels, suggesting that renin secretion increased as blood glucose levels deteriorated [19]. Durvasula RV et al. observed that exposure to high blood glucose levels increased renin activity, resulting in a 2.1-fold rise in angiotensin II (AII) [20]. AII further exacerbated progressive podocyte damage and loss in diabetic kidney disease [20]. Conversely, Fernandez-Cruz A et al. discovered that diabetic patients with nephropathy were more prone to low renin activity [21]. Two systematic studies of plasma renin activity, conducted on a substantial community sample of hypertensive patients (n = 1660 [22] and n = 4170 [23], respectively), revealed a wide distribution of activity level, particularly among diabetic patients. The RAAS system seemed to be influenced by the course of diabetes and its complications [13]. Integrating renin assessment into the diagnostic evaluation of T2DM patients requires clinicians to conduct a comprehensive judgment based on symptoms, medication adherence, and clinical manifestations. Accurate measurement of renin levels not only helps evaluate patient compliance with prescribed therapies but also assists clinicians in optimizing treatment strategies [19,24].

Researches have shown significant advancements in understanding the occurrence, progression, and management of T2DM in relation to the RAAS [9,19,22]. However, previous studies have predominantly concentrated on T2DM comorbid with hypertension or have been limited in scope, neglecting to track the quantitative or activity distribution of the RAAS throughout the course of the disease. With the progression of the disease and the treatment of hyperglycemia and hypertension, the activity or concentration of the RAAS may fluctuate, potentially leading to the development of drug resistance. Static analysis of RAAS at a certain stage may not fully elucidate its role in the pathogenesis and progression of the disease. Consequently, this cross-sectional study was undertaken to investigate the variations in RAAS within subgroups of complications and their correlation with glucose metabolism. The aim was to accurately assess the RAAS status of patients and provide guidance for precise clinical interventions.

## 2. Methods

### 2.1. Study subjects

The research involved a retrospective selection of 191 patients diagnosed with T2DM at the First People's Hospital of Pingdingshan between April 2023 and May 2024. Among them, 151 patients were diagnosed with T2DM and essential hypertension (DMHT), comprising 93 males and 58 females, with an average age of 52.30 years and a median diabetic duration of 48.0 months. Among 40 patients with T2DM and normotension (DMNT), there were 21 males and 19 females, with an average age of 46.58 years and a median diabetic duration of 24.0 months. A cohort of 46 healthy individuals was selected as the control group, all of whom attended our hospital for routine physical examinations during the same period. The control individuals had never suffered from diabetes or hypertension and had no family history of these conditions. The average age of the cohort was 50.63 years, with an equal distribution of males and females. The flow diagram of patient selection was shown in Fig 1. The data were accessed for research purposes on May 6, 2024. During or after the data collection, personal identifiers such as names were not utilized; instead, a numerical identifier was employed to anonymize the information. The data collected was intended solely for academic research purposes, and all personal information was treated with the utmost confidentiality. Written informed consent for participation was obtained from the patients.

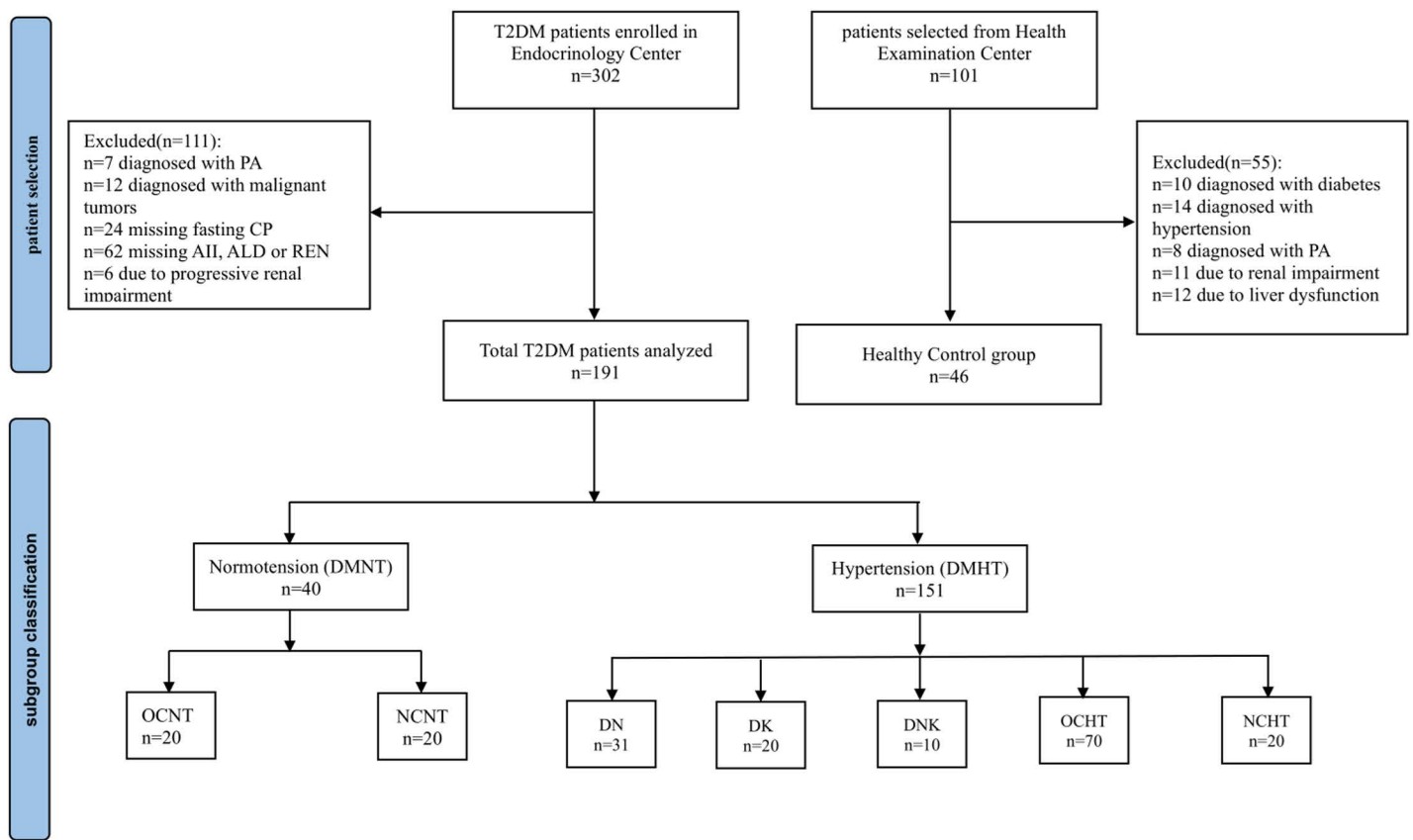

**Fig 1. Flow diagram of patient selection and subgroup classification.** T2DM, type 2 diabetes mellitus; PA, primary aldosteronism; DN, diabetic nephropathy; DK, diabetic ketoacidosis; DNK, diabetic nephropathy with ketoacidosis; OCHT, other diabetic complications in hypertensive patients; NCHT, no complications in hypertensive patients; OCNT, other diabetic complications in normotensive patients; NCNT, no complications in normotensive patients.

## 2.2. Group and subgroup

According to the diagnostic criteria for diabetic complications, a total of 151 patients with T2DM and hypertension (DMHT, n = 151) were classified into different subgroups according to their specific complications. These subgroups included those with diabetic nephropathy (DN, n = 31), diabetic ketoacidosis (DK, n = 20), diabetic nephropathy and ketoacidosis (DNK, n = 10), and other complications (OCHT, n = 70) such as peripheral vascular disease, neuropathy, retinopathy, or coronary heart disease. Additionally, there were 20 patients in the subgroup (NCHT, n = 20) without any complications. Furthermore, 40 patients with T2DM and normotension (DMNT, n = 40) were divided into subgroups based on the presence of other complications (OCNT, n = 20) or the absence of complications (NCNT, n = 20). The flow diagram of subgroup classification was shown in Fig 1.

## 2.3. Inclusion criteria and exclusion criteria

The study's inclusion criteria were as follows: (1) Type 2 Diabetes Mellitus (T2DM) diagnosed in accordance with the 1999 World Health Organization criteria [25]; (2) Essential hypertension defined as systolic blood pressure (SBP) ≥ 140 mmHg (1 mmHg = 0.133 kPa) and/or diastolic blood pressure (DBP) ≥ 90 mmHg, or a previous diagnosis or history of antihypertensive medication use; (3) Participants aged between 18 and 80 years; (4) Normal thyroid function.

Exclusion criteria included: (1) Secondary hypertension, severe liver dysfunction, stage IV or V chronic kidney disease, renal replacement therapy or transplantation, congestive heart failure, and malignant tumors; (2) Patients with primary aldosteronism; (3) Patients with Cushing syndrome; (4) Hyperparathyroidism; (5) Pregnant or breastfeeding individuals; (6) Individuals undergoing treatment with aldosterone receptor antagonists, direct renin inhibition therapy, oral contraceptives, or hormonal drug therapy(Aldosterone receptor antagonists and direct renin inhibitors significantly affect aldosterone and renin levels, subsequently influencing the aldosterone-to-renin ratio [26]). This study protected patient privacy by utilizing anonymized data, received ethical approval from the Ethics Committee of the First People's Hospital of Pingdingshan (PYLL2023022301), and adhered to the principles of the Declaration of Helsinki.

## 2.4. Method of detection

The patient's posture before blood sampling is crucial for measuring RAAS activity, and the aldosterone-to-renin ratio (ARR) test is most sensitive when samples are collected in the morning after patients have been up (sitting, standing, or walking) for at least 2 hours and seated for 10 minutes [26]. Consequently, all patients in this study adhered to this criterion for blood collection. Plasma renin (REN), angiotensin II (AII), and aldosterone (ALD) concentrations were quantified using AutoLumo A2000 Plus chemiluminescence analyzer (Autobio, China). The ARR was calculated using the formula ARR $= $ ALD/(REN $\times$ 10). Fasting blood glucose (FBG), fasting C-peptide (CP), triglycerides (TG), total cholesterol (TCHO), high-density lipoprotein cholesterol (HDL), low-density lipoprotein cholesterol (LDL), urea nitrogen (Urea), creatinine (Crea), potassium (K), sodium (Na), calcium (Ca), phosphorus (P), and other biochemical markers were analyzed using the Roche Cobas 8000 automated biochemical immunoassay analyzer. Homeostasis Model Assessment of β-cell function (HOMA-β) and Homeostasis Model Assessment of Insulin Resistance (HOMA-IR) was calculated by utilizing CP and FBG levels through the HOMA calculator available at http://www.dtu.ox.ac.uk/homacalculator/.

The random urinary albumin-to-creatinine ratio (UACR) was measured using the Spain Biosystems BA400 automatic specific protein analyzer. HbA1c levels were assessed using the Shanghai Huizhong MQ-6000 analyzer. Systolic blood pressure (SBP) and diastolic blood pressure (DBP) of the right upper limb were recorded after a 10-minute seated rest, with height (m) and weight (kg) standardized for measurement, and body mass index (BMI) calculated accordingly.

## 2.5. Statistical analysis

Statistical analysis was conducted using SPSS 21.0 software. Normally distributed continuous variables were presented as mean ± standard deviation, and non-normally distributed continuous variables were described as median with interquartile range. One-way ANOVA and Kruskal-Wallis *H* test were respectively utilized for comparing multiple groups based on normally or non-normally distributed continuous variables, and the Bonferroni test was employed for pairwise comparisons between groups. Spearman correlation analysis was employed to investigate factors influencing the RAAS system. Variables that followed non-normal distribution (AII, ALD, REN, ARR, UACR, CP, HOMA-β, HOMA-IR) were log-transformed. Multiple linear regression analysis was then utilized to identify independent factors influencing the RAAS system in four covariate models. Statistical significance was set at $p < 0.05$.

## 3. Results

### 3.1. General characterization of T2DM patients and healthy controls

Compared with the healthy controls, patients with T2DM and normotension (DMNT) exhibited higher HbA1c levels, higher FBG levels and lower ALD levels; Patients with T2DM

and hypertension (DMHT) had increased BMI, AII, HbA1c, FBG, SBP, DBP, and UACR levels, and decreased HDL levels. Furthermore, the age, BMI, AII, ALD, SBP, DBP, and UACR levels in hypertensive T2DM patients were significantly higher compared with normotensive patients ($p < 0.05$). No significant differences were observed in the remaining indicators among the three groups ($p > 0.05$) (Table 1).

## 3.2. Complication subgroup analysis of the RAAS system

The results revealed significant differences in AII, ALD, REN, and ARR levels across various complication subgroups ($p < 0.05$, Table 2). Among the DN, DK, DNK, OCHT, NCHT, OCNT, and NCNT subgroups, the concentrations of ALD and REN exhibited a consistent trend, while ARR showed an opposite trend to the REN concentration (Fig 2). This finding was consistent with the outcomes presented in Fig 5 subsequently, indicating a positive correlation between ALD and REN, and a negative correlation between ARR and REN. REN levels in DK/

**Table 1. General characterization of T2DM patients and healthy controls.**

| Variables | DMHT (n = 151) | DMNT (n = 40) | Controls (n = 46) | p value |
|---|---|---|---|---|
| Age (y) | 52.30 ± 12.86 | 46.58 ± 7.93 | 50.63 ± 12.58 | 0.030 |
| BMI (kg/m²) | 27.51 ± 3.64 | 25.92 ± 3.26 | 25.02 ± 2.33 | <0.001 |
| HbA1c (%) | 9.56 ± 2.15 | 9.22 ± 1.42 | 5.20 ± 0.39 | <0.001 |
| FBG (mmol/L) | 9.64 ± 2.83 | 8.95 ± 2.46 | 5.16 ± 0.40 | <0.001 |
| TCHO (mmol/L) | 4.81 ± 1.12 | 5.02 ± 1.65 | 4.77 ± 0.77 | 0.526 |
| TG (mmol/L) | 2.46 ± 1.75 | 2.47 ± 2.27 | 1.77 ± 1.13 | 0.058 |
| HDL (mmol/L) | 1.09 ± 0.26 | 1.16 ± 0.32 | 1.28 ± 0.30 | 0.001 |
| LDL (mmol/L) | 2.86 ± 0.92 | 2.93 ± 1.11 | 2.83 ± 0.76 | 0.882 |
| Urea (mmol/L) | 5.32 ± 1.99 | 5.09 ± 1.42 | 4.94 ± 1.50 | 0.439 |
| Crea (μmol/L) | 59.63 ± 31.29 | 58.53 ± 14.90 | 60.89 ± 15.41 | 0.918 |
| K (mmol/L) | 3.92 ± 0.45 | 4.00 ± 0.55 | 4.09 ± 0.38 | 0.081 |
| Na (mmol/L) | 139.90 ± 2.58 | 140.03 ± 3.36 | 140.67 ± 2.18 | 0.226 |
| Ca (mmol/L) | 2.30 ± 0.11 | 2.34 ± 0.11 | 2.31 ± 0.09 | 0.095 |
| P (mmol/L) | 1.150 ± 0.19 | 1.146 ± 0.22 | 1.142 ± 0.13 | 0.967 |
| SBP (mmHg) | 153.54 ± 19.54 | 123.88 ± 6.34 | 117.63 ± 5.59 | <0.001 |
| DBP (mmHg) | 93.35 ± 14.49 | 84.05 ± 4.56 | 79.11 ± 6.88 | <0.001 |
| Diabetic duration (M) | 48.0(6.0 ~ 120.0) | 24.0(4.3 ~ 60.0) | NA | 0.106 |
| AII (ng/L) | 115(106 ~ 125) | 101(96 ~ 112) | 105(96 ~ 115) | <0.001 |
| ALD (ng/L) | 149(116 ~ 190) | 119(97 ~ 150) | 163(142 ~ 209) | <0.001 |
| REN (ng/L) | 12.0(6.0 ~ 26.3) | 8.9(4.8 ~ 22.2) | 14.7(7.1 ~ 23.5) | 0.355 |
| ARR | 1.20(0.63 ~ 2.16) | 0.97(0.74 ~ 2.46) | 1.24(0.75 ~ 2.55) | 0.884 |
| UACR | 16.40(8.60 ~ 75.42) | 10.95(4.87 ~ 20.57) | 8.95(5.67 ~ 14.77) | 0.000 |
| CP (nmol/L) | 0.70(0.43 ~ 1.02) | 0.77(0.62 ~ 0.91) | NA | 0.428 |
| HOMA-β (%) | 41.6(26.7 ~ 60.0) | 51.5(35.1 ~ 67.8) | NA | 0.057 |
| HOMA-IR | 1.91(1.19 ~ 2.87) | 1.97(1.53 ~ 2.46) | NA | 0.711 |

Data are expressed as "mean ± standard deviation" or "median with interquartile range". NA, not applicable; DMHT, diabetes mellitus patients with hypertension; DMNT, diabetes mellitus patients with normotension; BMI, body mass index; HbA1c, Hemoglobin A1c; FBG, fast blood glucose; TCHO, total cholesterol; TG, triglycerides; HDL, high-density lipoprotein cholesterol; LDL, low-density lipoprotein cholesterol; Urea, urea nitrogen; Crea, creatinine; K, potassium; Na, sodium; Ca, calcium; P, phosphorus; SBP, systolic blood pressure; DBP, diastolic blood pressure; AII, angiotensin II; ALD, aldosterone; REN, renin; ARR, aldosterone-to-renin ratio; UACR, urinary albumin-to-creatinine ratio; CP, C-peptide; HOMA-β, Homeostasis Model Assessment of β-cell function; HOMA-IR, Homeostasis Model Assessment of insulin resistance.

Table 2. Complication subgroup analysis of the RAAS system in T2DM patients.

| Group and subgroup | AII (ng/L) | ALD (ng/L) | REN (ng/L) | ARR |
|---|---|---|---|---|
| DMHT(n = 151) | | | | |
| DN (n = 31) | 122 (110 ~ 129) | 155 (110 ~ 192) | 8.4 (4.7 ~ 15.6) | 1.70 (0.90 ~ 3.20) |
| DK (n = 20) | 117 (106 ~ 129) | 152 (113 ~ 201) | 16.2 (9.5 ~ 43.1) | 0.68 (0.41 ~ 1.59) |
| DNK (n = 10) | 120 (117 ~ 125) | 134 (115 ~ 206) | 6.8 (4.2 ~ 18.7) | 1.87 (1.20 ~ 2.76) |
| OCHT (n = 70) | 113 (106 ~ 124) | 147 (116 ~ 183) | 12.6 (5.9 ~ 26.7) | 1.10 (0.62 ~ 2.28) |
| NCHT (n = 20) | 104 (97 ~ 116) | 161 (132 ~ 241) | 19.3 (8.9 ~ 29.4) | 1.02 (0.59 ~ 2.78) |
| DMNT (n = 40) | | | | |
| OCNT (n = 20) | 96 (86 ~ 101) | 102 (97 ~ 128) | 5.1 (4.3 ~ 8.9) | 2.12 (1.11 ~ 2.50) |
| NCNT (n = 20) | 104 (101 ~ 113) | 146 (108 ~ 181) | 21.8 (8.3 ~ 29.6) | 0.80 (0.60 ~ 0.94) |
| Controls (n = 46) | 105 (96 ~ 115) | 163 (142 ~ 209) | 14.7 (7.1 ~ 23.5) | 1.24 (0.75 ~ 2.55) |
| H value | 46.859 | 30.375 | 23.988 | 20.811 |
| p value | <0.001 | <0.001 | 0.001 | 0.004 |

Data are expressed as median with interquartile range. DMHT, diabetes mellitus patients with hypertension; DMNT, diabetes mellitus patients with normotension; AII, angiotensin II; ALD, aldosterone; REN, renin; ARR, aldosterone-to renin-ratio; DN, diabetic nephropathy; DK, diabetic ketoacidosis; DNK, diabetic nephropathy with ketoacidosis; OCHT, other diabetic complications in hypertensive patients; NCHT, no complications in hypertensive patients; OCNT, other diabetic complications in normotensive patients; NCNT, no complications in normotensive patients.

control group were higher than those in DN/DNK group, and ARR increased sequentially in DK, control, and DN/DNK groups (Fig 2). REN and ALD levels decreased gradually in the NCHT/NCNT, OCHT, and OCNT groups, while ARR increased sequentially in these groups (Fig 2). This reveals that REN and ALD levels were correlated with hypertension, nephropathy, ketoacidosis, and other diabetic complications.

The highest level of AII was observed in DN group, while the lowest level was found in OCNT group compared with the other groups ($p < 0.001$). DN, DNK, and DK groups exhibited higher AII levels compared to both OCNT and control groups. The level of AII decreased successively in OCHT, NCHT/NCNT/control, and OCNT groups, indicating that AII levels are closely associated not only with hypertension but also with other diabetic complications (Fig 2).

### 3.3. Antihypertensive treatment of T2DM patients with hypertension

Among 151 hypertensive patients, the percentages of individuals using different types of anti-hypertensive medications were as follows: angiotensin-converting enzyme inhibitors/angiotensin receptor blockers (ACEI/ARBs) at 72.8%, calcium channel blockers (CCBs) at 74.8%, β-blockers at 29.8%, thiazide diuretics (TDs) at 13.9%, and α-blockers at 7.9%. The proportion of patients using one, two, three, and four drugs in combination was 35.1%, 37.1%, 20.5%, and 7.3%, respectively (Fig 3 and S1 Table). In each subgroup, over 60% of patients were prescribed ACEI/ARBs and CCBs, while 55% to 90% of patients received antihypertensive treatment either as monotherapy or in combination with two medications (Fig 3 and S1 Table).

By analyzing the effects of antihypertensive medications on glucose metabolism and RAAS (S2 Table), we found that β-blocker users exhibited decreased REN levels and increased ARR compared to non-users in both the overall hypertensive patient cohort (HT) and the subgroups with other complications (OCHT) and nephropathy (DN) (REN: 11.0 vs 13.0, 11.1 vs 16.6, 5.3 vs 12.9 ng/L, $p = 0.010, 0.036, 0.007$, respectively; ARR: 1.70 vs 1.07, 1.55 vs 1.00, 2.26 vs 1.18, $p = 0.004, 0.013, 0.01$, respectively; Fig 4). This finding was consistent with the multiple linear regression analysis, which showed a negative correlation between REN levels and β-blocker usage, and a positive correlation between ARR and β-blocker usage, as presented in Table 3.

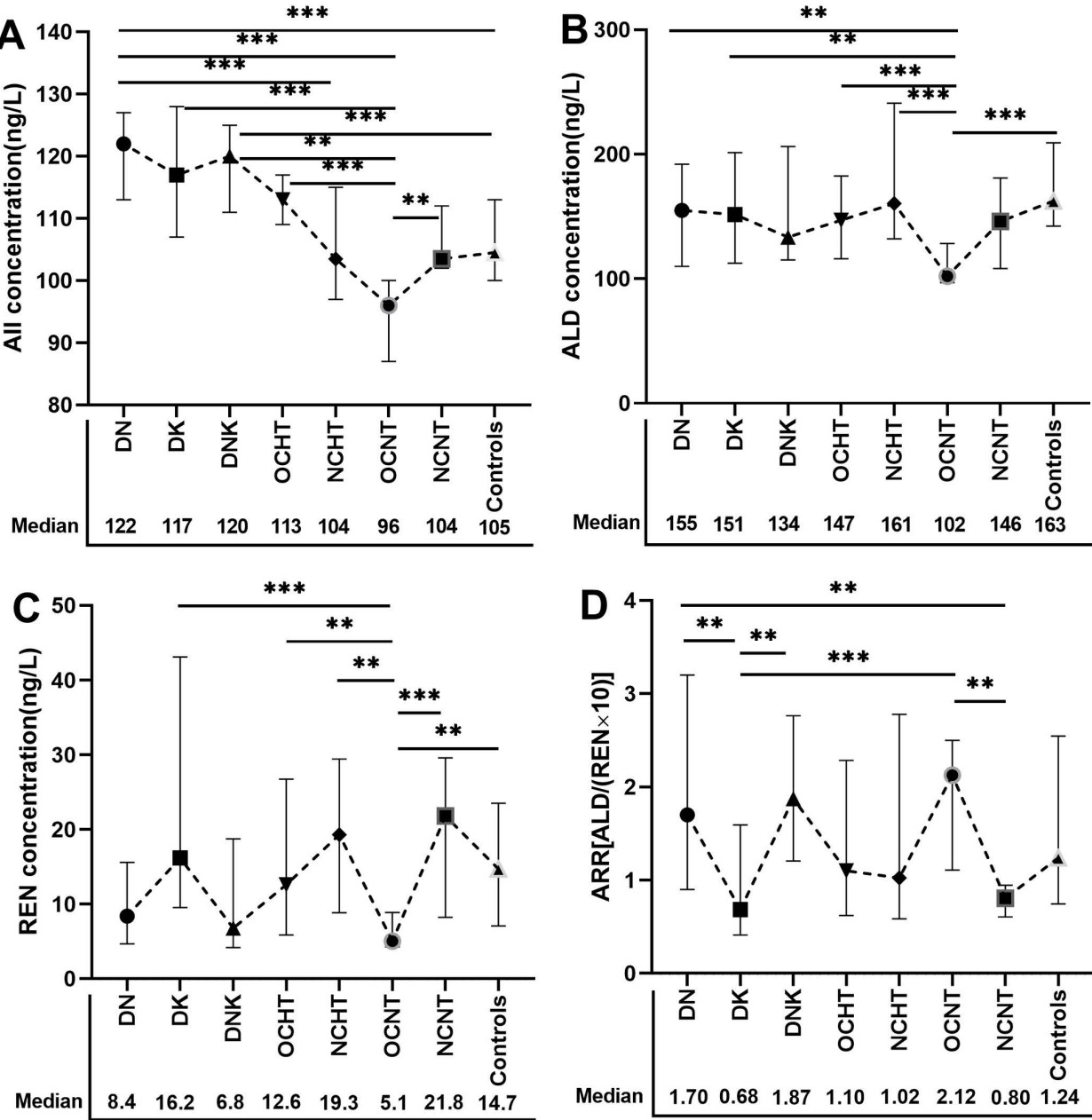

**Fig 2. The values of angiotensin II (A), aldosterone (B), renin(C), and aldosterone-to-renin ratio (D) varied among different complication subgroups.** DN, diabetic nephropathy; DK, diabetic ketoacidosis; DNK, diabetic nephropathy with ketoacidosis; OCHT, other diabetic complications in hypertensive patients; NCHT, no complications in hypertensive patients; OCNT, other diabetic complications in normotensive patients; NCNT, no complications in normotensive patients. (*$p < 0.05$, **$p < 0.01$, *** $p < 0.001$).

### 3.4. Antidiabetic treatment of T2DM patients and its influence on RAAS

S3 Table outlined the glucose-lowering medications that patients were taking across various subgroups. All T2DM patients (n = 191) received antidiabetic treatment, among which 95 (49.7%) took SGLT-2is, 40 (20.9%) took GLP-1RAs injection and 55(28.8%) took DPP-4is. SGLT-2is, GLP-1RAs, DPP-4is and other glucose-lowering agents had no statistically significant effect on the RAAS system ($p > 0.05$) as shown in S4 Table.

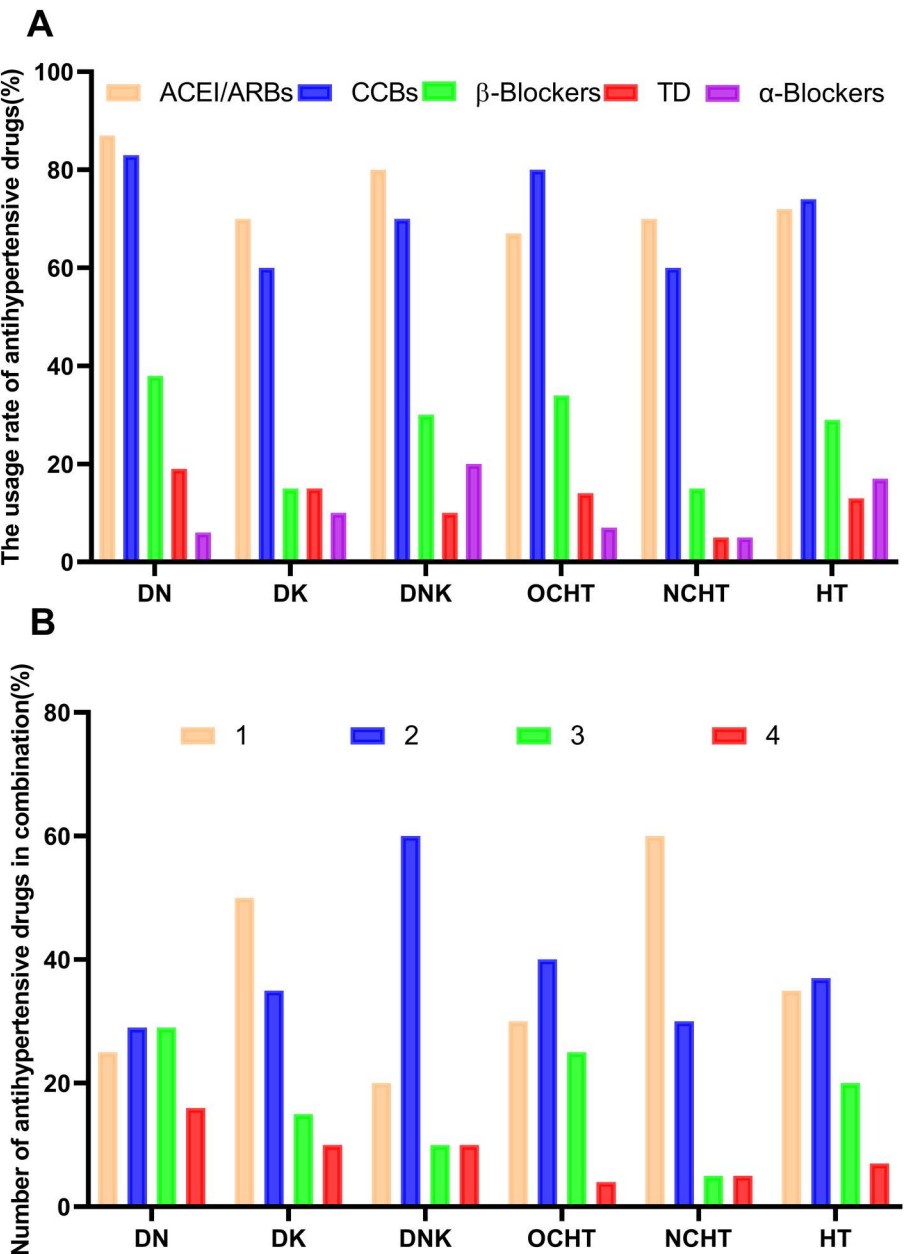

**Fig 3. Antihypertensive treatment of 151 T2DM patients with hypertension at different subgroups. (A)**The usage rate of antihypertensive drugs at different subgroups; **(B)** Number of antihypertensive drugs in combination at different subgroups. DN, diabetic nephropathy; DK, diabetic ketoacidosis; DNK, diabetic nephropathy with ketoacidosis; OCHT, other diabetic complications in hypertensive patients; NCHT, no complications in hypertensive patients; HT, hypertension; ACEI/ARBs, angiotensin-converting enzyme inhibitors/angiotensin receptor blockers; CCBs, calcium channel blockers; TDs, thiazide diuretics.

### 3.5. Spearman correlation matrix analysis of the RAAS system and glucose metabolism indexes

To explore their interrelationships further, a correlation matrix analysis was conducted among 10 variables, including AII, ALD, REN, ARR, HbA1c, FBG, CP, HOMA-β, HOMA-IR, and UACR. The correlation heatmap was presented in Fig 5.

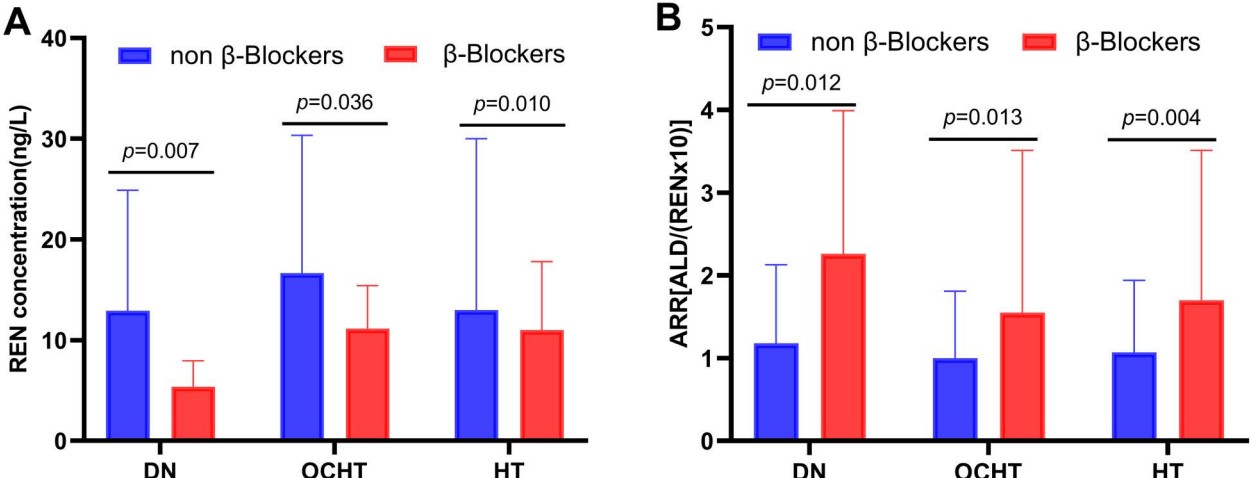

**Fig 4. The effect of β-blocker treatment on renin (A) and aldosterone-to-renin ratio (B).** DN, diabetic nephropathy; OCHT, other diabetic complications in hypertensive patients; HT, hypertension.

In the hypertensive (HT) group (Fig 5A), ALD, REN, CP, HOMA-β, and HOMA-IR exhibited positive correlations with each other. REN showed a negative correlation with UACR. ARR was negatively correlated with REN and positively correlated with UACR. FBG was positively correlated with HbA1c and HOMA-IR, while displaying negative correlations with HOMA-β.

In the normotensive (NT) group (Fig 5B), ALD, CP, HOMA-β, HOMA-IR, and UACR showed positive correlations with each other. AII, ALD, REN, CP, and HOMA-IR also exhibited positive correlations with each other. REN was also positively correlated with HbA1c and FBG. ARR displayed negative correlations with REN, AII, HbA1c, and FBG, while showing positive correlations with HOMA-β and UACR. FBG was positively correlated with HbA1c, REN, and HOMA-IR, and negatively correlated with ARR and HOMA-β.

Correlation analyses at HT and NT subgroups were displayed in S1 and S2 Figs. Consistent with the HT group above, ALD, REN, CP, HOMA-β and HOMA-IR exhibited positive correlations with each other at DN, DK, DNK, OCHT and NCHT subgroups. In addition, REN, ALD and AII were negatively correlated with HbA1c and FBG at DNK group (S1 Fig). As shown in S2 Fig, ALD exhibited positive correlations with CP and HOMA-IR, while REN displayed positive correlations with HbA1c, FBG and HOMA-IR at OCNT and NCNT subgroups.

## 3.6. Multiple linear regression analysis of the RAAS system and glucose metabolism indexes

Multiple linear regression analysis was performed with AII, ALD, REN, and ARR as dependent variables, and factors correlated with RAAS levels as independent variables in four covariate models. The results of each group were presented in Table 3.

In the hypertensive (HT) group, it was observed that ALD levels increased with higher levels of CP and HOMA-IR, and this correlation remained significant even after adjusting for REN, AII, antidiabetic and antihypertensive treatments. This suggested that the correlation was not influenced by the RAAS system or by anti-RAAS and antidiabetic treatments. REN levels were found to decrease with increasing UACR and β-blocker use, independently of the RAAS system and antidiabetic treatment but not independent of antihypertensive therapy. Additionally, ARR values were noted to increase with higher UACR and β-blocker usage,

**Table 3. Multiple linear regression analysis of the RAAS system and glucose metabolism indexes in HT and NT patients.**

| HT—Linear regression β coefficients | | | | | |
|---|---|---|---|---|---|
| Dependent variables | Independent variables | Model 1 | Model 2 | Model 3 | Model 4 |
| Log-ALD | Log-CP | 0.228 (0.130 ~ 0.325) * | 0.206 (0.123 ~ 0.288) * | 0.235(0.133 ~ 0.338) * | 0.230 (0.129 ~ 0.332) * |
| | Log-HOMA-IR | 0.221 (0.128 ~ 0.315) * | 0.202 (0.123 ~ 0.281) * | 0.227 (0.130 ~ 0.325) * | 0.223 (0.126 ~ 0.320) * |
| | Log-HOMA-β | 0.105 (-0.004 ~ 0.213) | 0.087 (-0.007 ~ 0.180) | 0.122 (-0.004 ~ 0.248) | 0.112 (-0.001 ~ 0.224) |
| Log-REN | Log-UACR | -0.139 (-0.246 ~ -0.031) * | -0.139 (-0.229 ~ -0.026) * | -0.136 (-0.251 ~ -0.021) * | -0.134 (-0.244 ~ -0.023) * |
| | Log-CP | 0.102 (-0.155 ~ 0.359) | -0.224 (-0.455 ~ 0.008) | 0.137 (-0.137 ~ 0.411) | 0.071 (-0.186 ~ 0.328) |
| | Log- HOMA-β | 0.078 (-0.192 ~ 0.348) | -0.046 (-0.278 ~ 0.186) | 0.188 (-0.129 ~ 0.504) | 0.062 (-0.209 ~ 0.333) |
| | β-Blockers | -0.169 (-0.311 ~ -0.026) * | -0.164 (-0.285 ~ -0.043) * | -0.173 (-0.323 ~ -0.023) * | -0.027 (-0.819 ~ 0.765) |
| Log-ARR | Log-UACR | 0.139 (0.047 ~ 0.230) * | 0.003 (-0.003 ~ 0.008) | 0.139 (0.042 ~ 0.237) * | 0.139 (0.046 ~ 0.233) * |
| | β-Blockers | 0.154 (0.031 ~ 0.277) * | 0.002 (-0.005 ~ 0.009) | 0.157 (0.030 ~ 0.285) * | 0.049 (-0.632 ~ 0.731) |
| NT—Linear regression β coefficients | | | | | |
| Dependent variables | Independent variables | Model 1 | Model 2 | Model 3 | |
| Log-ALD | Log-CP | 0.644 (0.426 ~ 0.863) * | 0.531 (0.291 ~ 0.770) * | 0.665 (0.399 ~ 0.932) * | |
| | Log-HOMA-IR | 0.543 (0.315 ~ 0.770) * | 0.473 (0.209 ~ 0.737) * | 0.554 (0.293 ~ 0.815) * | |
| | Log-UACR | 0.129 (-0.041 ~ 0.298) | 0.220 (0.101 ~ 0.339) * | 0.034 (-0.198 ~ 0.266) | |
| Log-REN | HbA1c | 0.117 (0.019 ~ 0.215) * | 0.113 (0.035 ~ 0.190) * | 0.157 (0.013 ~ 0.301) * | |
| | FBG | 0.092 (0.038 ~ 0.145) * | 0.102 (0.071 ~ 0.134) * | 0.125 (0.055 ~ 0.195) * | |
| | Log-HOMA-IR | 1.198 (0.516 ~ 1.880) * | 1.006 (0.078 ~ 1.934) * | 1.601 (0.581 ~ 2.621) * | |
| Log-ARR | HbA1c | -0.115 (-0.197 ~ -0.033) * | -0.001 (-0.002 ~ 0.000) | -0.142 (-0.265 ~ -0.019) * | |
| | FBG | -0.095 (-0.135 ~ -0.054) * | 0.001 (-0.001 ~ 0.001) | -0.120 (-0.173 ~ -0.066) * | |
| | Log-UACR | 0.470 (0.117 ~ 0.822) * | 0.007 (0.000 ~ 0.013) * | 0.695 (0.150 ~ 1.241) * | |
| | Log-HOMA-β | 0.557 (0.032 ~ 1.081) * | 0.011 (0.001 ~ 0.021) * | 0.700 (-0.124 ~ 1.524) | |
| Log-AII | Log-CP | 0.206 (0.017 ~ 0.396) * | -0.048 (-0.310 ~ 0.214) | 0.295 (0.014 ~ 0.515) * | |
| | Log-HOMA-IR | 0.171 (-0.008 ~ 0.349) | -0.110 (-0.340 ~ 0.120) | 0.229 (-0.029 ~ 0.486) | |

Model 1: age, total cholesterol, triglycerides, high-density lipoprotein cholesterol, low-density lipoprotein cholesterol, urea nitrogen, creatinine, potassium, sodium, calcium, phosphorus, systolic blood pressure, diastolic blood pressure, body mass index and diabetic duration. Model 2(aldosterone analyses): Model 1 + log-renin and log-angiotensin II. Model 2(renin analyses): Model 1 + log-aldosterone and log-angiotensin II. Model 2 (ARR analyses): Model 1 + log-aldosterone, log-renin and log-angiotensin II. Model 2(AII analyses): Model 1 + log-aldosterone and log-renin. Model 3: Model 1 + metformin, α-glucosidase inhibitors, insulin, sodium-glucose cotransporter-2 inhibitors, glucagon-like peptide-1 receptor agonists, dipeptidyl peptidase-4 inhibitors, thiazolidinediones, sulfonylureas, and glinides. Model 4: Model 1 + NO. of drugs, angiotensin-converting enzyme inhibitors/angiotensin receptor blockers, calcium channel blockers, β-blockers, thiazide diuretics and α-blockers. ALD, aldosterone; REN, renin; ARR, aldosterone-to-renin ratio; AII, angiotensin II; HbA1c, Hemoglobin A1c; FBG, fast blood glucose; UACR, urinary albumin-to-creatinine ratio; CP, C-peptide; HOMA-IR, Homeostasis Model Assessment of insulin resistance; HOMA-β, Homeostasis Model Assessment of β-cell function. (*$p < 0.05$)

independently of antidiabetic therapy, although this correlation was not independent of the RAAS system.

In the normotensive (NT) group, it was observed that ALD levels increased with higher CP and HOMA-IR, and this correlation was independent of the RAAS system and anti-diabetic treatment. REN levels were found to increase with higher levels of HbA1c, FBG, and HOMA-IR, independently of the RAAS system and antidiabetic therapy. ARR values increased with higher UACR and HOMA-β levels, and this correlation was also independent of the RAAS system. However, ARR values decreased with higher HbA1c and FBG levels, but this correlation was not independent of the RAAS system.

## 4. Discussion

Type 2 diabetes mellitus (T2DM) patients are often accompanied by hypertension and various acute or chronic complications that may activate or inhibit the renin-angiotensin-aldosterone

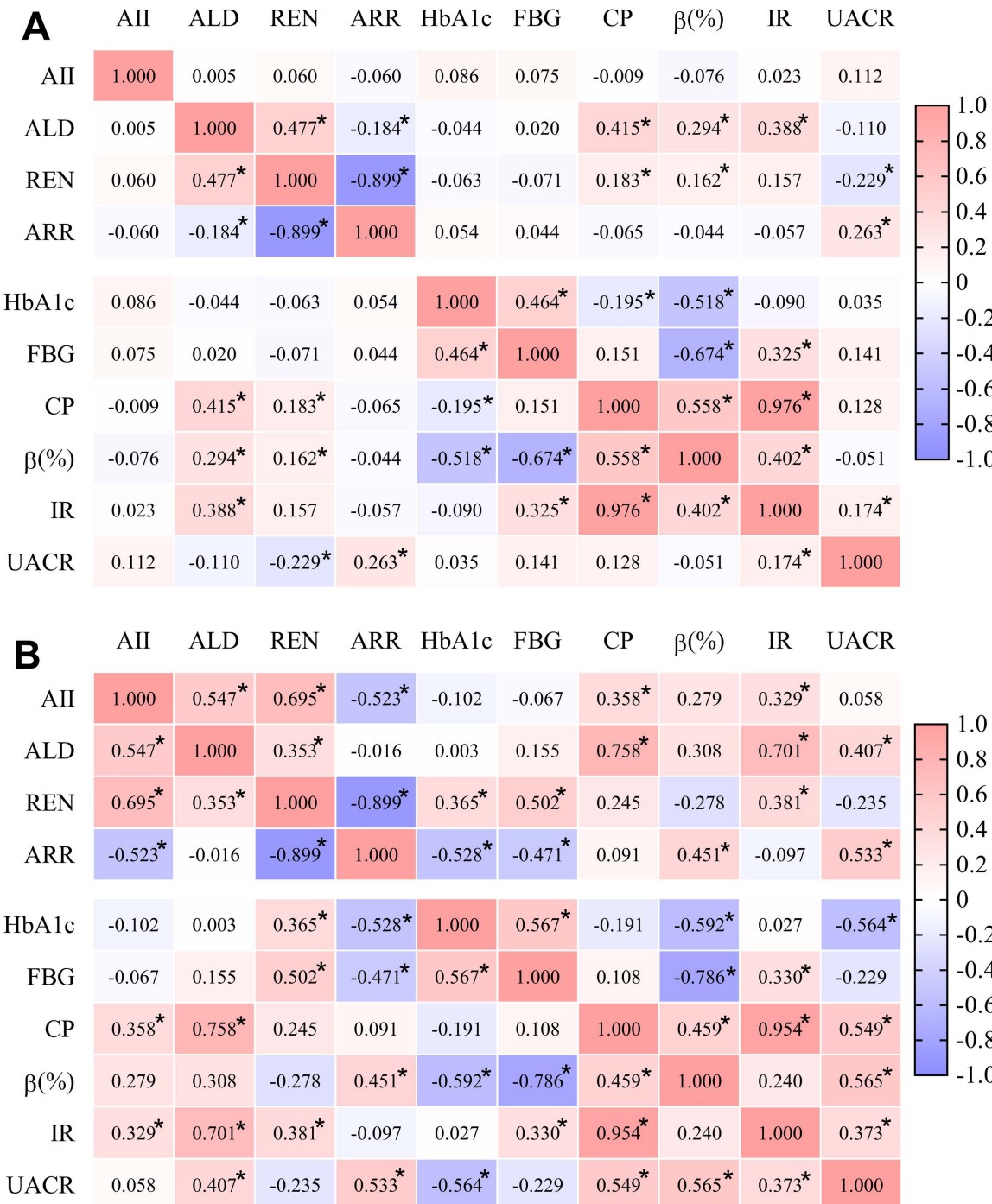

**Fig 5. Spearman correlation matrix analysis of the RAAS system and glucose metabolism indexes in hypertensive (A) and normotensive (B) patients.** AII, Angiotensin II; ALD, aldosterone; REN, renin; ARR, aldosterone-to-renin ratio; HbA1c, hemoglobin A1c; FBG, fast blood glucose; CP, C-peptide; β (%), Homeostatic Model Assessment of β-cell function; IR, Homeostatic Model Assessment of Insulin Resistance; UACR, urinary albumin-to-creatinine ratio. Correlation analyses at different complication subgroups were displayed in S1 and S2 Figs (*$p < 0.05$).

system (RAAS), leading to increased rates of hospitalization and mortality [12–15]. However, limited research has been conducted on the quantitative distribution of RAAS in relation to different diabetic complications and their association with glucose metabolism. Therefore, we investigated this issue in this cross-sectional study and found that the activation status of the RAAS system varied among T2DM patients with distinct complications, highlighting the importance of clinical differentiation. Notably, aldosterone and renin were positively associated with insulin resistance and hyperglycemia, and detrimentally impaired glucose metabolism. Blocking the RAAS system shows potential in improving insulin sensitivity and pancreatic β-cell function, thereby ameliorating abnormal glucose tolerance and metabolic disorders.

Renin, an enzymatic protein released by juxtaglomerular cells, catalyzes the conversion of angiotensinogen in the bloodstream to angiotensin I. This compound is further metabolized by angiotensin-converting enzyme into angiotensin II (AII) [10,13]. The binding of AII to angiotensin type I receptors triggers a cascade of physiological responses, including vasoconstriction, increased aldosterone secretion, enhanced sodium and water reabsorption in the kidneys, expansion of extracellular fluid volume, and elevation of blood pressure [10,13]. Our study found that ALD and REN levels exhibited a consistent trend, while ARR showed an opposite trend to REN levels, indicating that ALD secretion was primarily influenced by the RAAS system. The activation status of the RAAS system varied among T2DM patients with distinct complications, highlighting the importance of conducting a thorough clinical evaluation. REN level was higher in DK group in comparison to DN/DNK group. The variance is probably due to a decrease in extracellular fluid volume and renal blood flow caused by dehydration in DK, which stimulates juxtaglomerular cells and increases renin secretion [27]. Conversely, the absence of such a reaction in DNK indicates impaired juxtaglomerular cell function, leading to an insufficient increase in renin release, even when stimulated by dehydration [27]. REN and ALD levels decreased gradually in the NCHT/NCNT, OCHT, and OCNT groups, while ARR increased sequentially in these groups. This reveals that REN and ALD levels were not only correlated with hypertension, nephropathy, and ketoacidosis, but also associated with other diabetic complications such as peripheral vascular disease, neuropathy, retinopathy, and coronary heart disease. This phenomenon could be attributed to the hyalinization of the afferent arteriole and juxtaglomerular cells in individuals with diabetic nephropathy, leading to a decrease in renin secretion [22]. Additionally, diabetic autonomic neuropathy contributes to a reduction in catecholamine-induced stimulation of renin release [22].

AII, a peptide known for its vasoconstrictive properties, can lead to increased blood pressure. In this study, the OCNT group exhibited a simultaneous decrease in the concentrations of REN and AII in comparison to the control group. Conversely, the DN and DNK groups showed an increase in AII concentration, while the concentration of REN decreased in these groups. Patients with DN and hypertension were observed to be older, with a longer diabetic duration, and demonstrated poor glycemic control in comparison to individuals without these complications. The conflicting results indicated that AII may play a more significant role than renin in the regulation of blood pressure among DN patients [28,29]. The elevation of AII levels may be associated with a reduction in extracellular fluid volume or heightened sympathetic nerve activity resulting from prolonged poor blood glucose control, both of which can be ameliorated through effective blood glucose management [19,20,30]. The level of AII decreased successively in OCHT, NCHT/NCNT/control, and OCNT groups, indicating that AII levels were closely associated not only with hypertension but also with other diabetes-related complications [30]. AII plays a crucial role in the regulation of blood pressure and is the target of various antihypertensive medications like ACE inhibitors and ARBs. Consequently, there is a growing focus on the clinical detection of AII.

RAAS inhibitors (ACEI/ARBs), calcium channel blockers (CCBs), β-blockers, and thiazide diuretics (TDs) are commonly recommended antihypertensive medications for individuals with diabetes [17,18,31]. In various subgroups of hypertension, more than 60% of individuals were prescribed ACEI/ARBs and CCBs, with 15% to 38.7% receiving β-blockers. Additionally, 55% to 90% of patients were administered antihypertensive medications, either as monotherapy or in combination with two drugs. Griffin TP et al. found that β-blockers reduce renin levels while maintaining aldosterone levels, leading to an increase in ARR [32,33]. This observation aligns with our own findings. The potential explanation for this effect may lie in the β-blockers' ability to inhibit sympathetic nervous system activity, which in turn decreases renin secretion and ultimately raises the ARR [32,33]. Several studies have indicated that CCBs [34], α-blockers [34], and TDs [35] were associated with a slight increase in renin and a decrease in aldosterone and ARR. Zhao D et al. have suggested that CCBs treatment was negatively correlated with HbA1c levels in patients with T2DM and hypertension, and positively correlated with β-cell function [10]. However, CCBs and other antihypertensive medications did not demonstrate similar effects in our study. The influence of antihypertensive therapy on RAAS and glucose metabolism remains a topic of ongoing debate.

Sodium-glucose cotransporter-2 inhibitors (SGLT-2is), glucagon-like peptide-1 receptor agonists (GLP-1RAs), and dipeptidyl peptidase-4 inhibitors (DPP-4is) are glucose-lowering agents that have been shown to reduce cardiovascular risk in T2DM patients [36]. The RAAS may play a role in this mechanism. One study indicated that SGLT-2is did not influence ALD, REN levels, or ARR [37]. Conversely, another study observed an elevation in REN levels without a corresponding change in ALD after the use of SGLT-2is [38]. Further evidence indicated that SGLT-2is caused REN increase in early phase of treatment [39], whereas REN and ALD levels remain unchanged with long-term use [40]. GLP-1RAs have been shown to inhibit the synthesis of AII and enhance the inactivation of its circulating form, thereby mitigating the detrimental effects of AII on renal, cardiovascular, and other tissues [36]. Several studies have indicated that GLP-1 and RAAS antagonists exhibit synergistic effects in multiple tissues and organs. For instance, Miyagawa K et al. conducted experiments utilizing murine models that were exposed to a high-fat diet and diabetic conditions [41]. Their findings revealed that the combination of the DPP-4i vildagliptin and the ARB valsartan led to a significant reduction in blood glucose levels, enhanced insulin sensitivity, and decreased serum inflammatory cytokines compared to monotherapy [41]. However, our research indicated that SGLT-2is, GLP-1RAs, DPP-4is, and other glucose-lowering agents did not produce statistically significant effects on the RAAS system. The available data on the relationships between SGLT-2is, GLP-1RAs, and DPP-4is in relation to RAAS activation remain inconclusive and contradictory [38,39]. This heterogeneity may be attributed to various study designs (e.g., inclusion criteria, follow-up period), control conditions (such as self-control), dietary factors (including sodium and protein intake), diverse sampling protocols, multiple interfering medications at different doses, and varying adherence to prescribed treatments [36,38,39].

A correlation matrix analysis was performed to investigate the relationship between glucose metabolism and the RAAS system. The results revealed positive correlations among ALD, REN, CP, and HOMA-IR. Furthermore, multiple linear regression analysis indicated that ALD levels increased with higher levels of CP and HOMA-IR, independent of the RAAS system, antidiabetic therapy, anti-RAAS medications, and other antihypertensive treatments. Our study indicated that the levels of ALD were observed to fluctuate in correlation with REN levels. However, the correlation between ALD and glucose metabolism seemed to remain unaltered by REN and was not impacted by known risk factors for T2DM, such as age, BMI, lipid profiles, creatinine levels, and hypertension. Joseph JJ and colleagues' findings further corroborate our assertion, demonstrating a positive correlation between elevated aldosterone levels

and increased blood glucose levels, as well as insulin resistance [9]. Preclinical investigations have revealed that ALD can impede β-cell function and insulin secretion by inhibiting insulin signaling pathways and reducing glucose-stimulated insulin release [12,15]. It is noteworthy that aldosterone synthase-deficient mice exhibited a notable increase in glucose-stimulated insulin secretion [42], highlighting the importance of understanding the role of aldosterone in the pathophysiology and development of T2DM. Furthermore, a prospective study conducted in a Japanese cohort indicated that elevated blood aldosterone levels could serve as a predictor for the development of insulin resistance [43]. Clinical trials targeting diabetes prevention have demonstrated a reduced risk of diabetes in individuals treated with RAAS inhibitors such as ACEIs or ARBs [44,45].

In the hypertensive cohort, a negative correlation was noted between renin concentrations and urinary microalbumin excretion, as well as the utilization of β-blockers. This may be due to the hyalinization of the afferent arteriole and juxtaglomerular cells in individuals with elevated urinary microalbumin levels [22], and the inhibition of sympathetic nerve activity induced by β-blocker usage [32,33], leading to reduced renin synthesis and secretion by the kidneys. Conversely, individuals with normal blood pressure included in the study exhibited a shorter duration of illness, fewer complications, no progression to diabetic nephropathy, and no use of β-blockers. Consequently, in the normotensive cohort, renin levels were positively correlated with blood glucose levels, HbA1c, and HOMA-IR, independently of aldosterone. This finding is consistent with the outcomes of the Griffin TP study, which demonstrated a positive correlation between increased renin levels and higher HbA1c, indicating that renin levels increase as blood glucose levels deteriorates [19]. The hyperglycemic milieu induces prompt expression of the succinate receptor GPR91 in the kidney [46,47]. Activation of this receptor initiates paracrine signaling pathways, ultimately resulting in the release of renin [46,47]. Durvasula RV et al. reported that exposure to high glucose levels augmented renin activity, resulting in a 2.1-fold rise in the action of AII [20]. Subsequently, AII can exacerbate progressive podocyte damage and loss in diabetic kidney disease [20]. However, Fernandez-Cruz A et al. observed that diabetic patients with kidney disease were more likely to experience reduced renin activity [21]. Two systematic studies of plasma renin activity, conducted on a substantial community sample of hypertensive patients (n = 1660 [22] and n = 4170 [23], respectively), revealed a wide distribution of activity level, particularly among diabetic patients.

The investigations have demonstrated that the RAAS was influenced by the progression of diabetes, the presence of complications, and the level of blood glucose. This research examined the differences in renin levels among various subgroups of complications and identified diverse associations with glucose metabolism in individuals with hypertension in comparison to those with normal blood pressure. The integration of renin detection in the diagnostic evaluation and follow-up of patients with T2DM requires healthcare professionals to perform a thorough assessment considering symptoms, prescribed medications, and clinical manifestations. Precise measurement of renin levels not only aids in assessing patients' compliance with prescribed therapies but also assists clinicians in optimizing therapeutic strategies [19,24]. Individuals diagnosed with "low renin" hypertension may demonstrate tendencies towards sodium and water retention, indicating the potential benefit of utilizing antihypertensive medications like TDs and CCBs [19]. In contrast, patients with "moderate to high renin" hypertension may display excessive vasoconstriction attributed to renin-angiotensin II activity, thus justifying the prescription of ACEIs/ARBs and β-blockers as preferred treatment options [19].

The study has several potential limitations that should be acknowledged. Firstly, the research noted variations in the activation state of the RAAS system among T2DM patients with different complications. However, due to the limited sample size, the study did not

perform multiple linear regression analysis of the RAAS system and glucose metabolism within specific complication subgroups. Additionally, it did not analyze the influence of antihypertensive therapy on the RAAS system and glucose metabolism in the DK, DNK, and NCHT groups. Secondly, the analysis did not consider the dosage of antihypertensive medications. Thirdly, we did not change any antidiabetic drugs prior to the study. Additionally, the RAAS indicators were only measured at one single time point, failing to assess changes in ALD and REN with glucose metabolism. Consequently, further prospective studies designed with larger sample sizes to determine the role of these findings are warranted.

## 5. Conclusions

The activation status of the RAAS system varied among T2DM patients with different complications, underscoring the importance of clinical differentiation. Aldosterone and renin were positively associated with insulin resistance and hyperglycemia, and negatively impacted glucose metabolism. Inhibiting the RAAS system has the potential to enhance insulin sensitivity and pancreatic β-cell function, ultimately improving abnormal glucose tolerance and metabolic disorders.

## Supporting information

**S1 Table. Antihypertensive treatment of 151 T2DM patients with hypertension at different subgroups.**
(DOCX)

**S2 Table. The influence of antihypertensive therapy on the RAAS system and glucose metabolism indexes.**
(DOCX)

**S3 Table. Antidiabetic treatment of 191 T2DM patients with hypertension or normotension at different subgroups.**
(DOCX)

**S4 Table. The influence of antidiabetic therapy on the RAAS system.**
(DOCX)

**S1 Fig. Spearman correlation matrix analysis of the RAAS system and glucose metabolism indexes at different hypertension subgroups.** Heat-map with background in three-color scale where − 1 = blue, + 1 = red and 0 = white. DN, diabetic nephropathy; DK, diabetic ketoacidosis; DNK, diabetic nephropathy with ketoacidosis; OCHT, other diabetic complications in hypertensive patients; NCHT, no complications in hypertensive patients; AII, Angiotensin II; ALD, aldosterone; REN, renin; ARR, aldosterone-to-renin ratio; HbA1c, hemoglobin A1c; FBG, fast blood glucose; CP, C-peptide; β (%), Homeostatic Model Assessment of β-cell function; IR, Homeostatic Model Assessment of Insulin Resistance; UACR, urinary albumin-to-creatinine ratio. (*$p < 0.05$).
(TIF)

**S2 Fig. Spearman correlation matrix analysis of the RAAS system and glucose metabolism indexes at different normotension subgroups.** Heat-map with background in three-color scale where − 1 = blue, + 1 = red and 0 = white. OCNT, other diabetic complications in normotensive patients; NCNT, no complications in normotensive patients; AII, Angiotensin II; ALD, aldosterone; REN, renin; ARR, aldosterone-to-renin ratio; HbA1c, hemoglobin A1c; FBG, fast blood glucose; CP, C-peptide; β (%), Homeostatic Model Assessment of β-cell function; IR,

Homeostatic Model Assessment of Insulin Resistance; UACR, urinary albumin-to-creatinine ratio. (*$p < 0.05$).
(TIF)

## Acknowledgments

We would like to thank the subjects involved in this study and the staff of Department of Laboratory Medicine for their technical support.

## Author contributions

**Conceptualization:** Ningning Wang, Junfeng Kong, Baohong Yue.

**Data curation:** Ningning Wang, Junhui Li, Erjun Tian, Shutong Li.

**Formal analysis:** Ningning Wang, Erjun Tian.

**Funding acquisition:** Fei Cao.

**Investigation:** Ningning Wang, Shuai Liu, Junfeng Kong.

**Methodology:** Ningning Wang, Junhui Li, Shutong Li, Junfeng Kong.

**Project administration:** Fei Cao.

**Software:** Ningning Wang, Shutong Li, Shuai Liu.

**Supervision:** Erjun Tian, Junfeng Kong, Baohong Yue.

**Validation:** Shutong Li, Shuai Liu.

**Visualization:** Ningning Wang.

**Writing – original draft:** Ningning Wang.

**Writing – review & editing:** Ningning Wang, Junhui Li, Erjun Tian, Shutong Li, Shuai Liu, Fei Cao, Junfeng Kong, Baohong Yue.

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
