## [Decision Letter · Decision Letter 0]

4 Oct 2024

PONE-D-24-34834Renin-angiotensin-aldosterone system variations in type 2 diabetes mellitus patients with different complications: implications for glucose metabolismPLOS ONE

Dear Dr. Yue,

Thank you for submitting your manuscript to PLOS ONE. After careful consideration, we feel that it has merit but does not fully meet PLOS ONE’s publication criteria as it currently stands. Therefore, we invite you to submit a revised version of the manuscript that addresses the points raised during the review process.

**ACADEMIC EDITOR**

The authors provide a detailed narrative of RAAS activation both in T2DM patients as well as different conditions related to T2DM. Their data confirms that the activation status of the RAAS system differed among T2DM patients with different complications, highlighting the need for clinical differentiation.

**General comments**

There are several grammatical and punctuation errors throughout the manuscript, which can be revised with the expertise of a language editor.

**Please also address the following:**

Submission page: Corresponding Author; Order of Authors: The is abbreviated incorrectly. Please correct to abbreviated term “ph.D” to “PhD”; and “CHINA” to “China”.

**Abstract**

Change “Result” to “Results”; “cross” to “across”.

“Key words” should be one word, i.e., “Keywords”.

**Introduction**

• Page 1: “The global diabetic population has surged to 537 million and is projected to escalate to 642 million by 2040” - Does this statistic represent both Type 1 and Type 2 diabetes, or just Type 2 diabetes? Clarification on this point would help readers understand the scope of the study.

• Page 2-3: Revise the grammar or reword the sentence “China has witnessed a significant increase in diabetes prevalence in recent years, with the highest number of people with diabetes [3, 4].

• Please provide statistics to support this statement. Additionally, is this true for both Type 1 and Type 2 diabetes?

• The statement “Research indicates that the activation status of the renin-angiotensin aldosterone system (RAAS) undergoes alterations in diabetic patients with various complications [9-11]” Please specify if this is for Type 1 or Type 2 diabetic patients or inclusive of both types.

• Please change “rate” to “rates”.

• Please insert “levels” after “elevated HbA1c” .

• Please change “blood glucose control deteriorated” to “blood glucose levels deteriorated”

• The statement “A comprehensive study with a substantial sample size revealed a wide distribution of renin activity in hypertensive patients, particularly among those with diabetes [19].” Should be expanded to include sample size and more data to enhance the relevance

• Please add reference to support this statement “Accurate measurement of renin levels not only helps evaluate patient compliance with prescribed therapies but also assists clinicians in optimizing treatment strategies.”

**Method**

•Study subjects: Be consistent and use two decimal places for “average age of 52.3 years”

•Verbal consent is indicated, however written informed consent is most applicable, especially when dealing with personal and sensitive health data. Was verbal consent the only form of consent? If so, please clarify why this was deemed appropriate. Also, consider adding whether the study followed any privacy guidelines, such as anonymizing data.

•Why are individuals undergoing treatment with aldosterone receptor antagonists and direct renin inhibitors excluded? Are these treatments expected to interfere with RAAS measurements? A brief justification would clarify this decision so that other readers understand.

•The control group consists of “46 healthy individuals.” How was the absence of diabetes or hypertension verified? Were any potential confounders (such as family history of diabetes or hypertension) accounted for when selecting the control group?

•Regarding the statement “The individuals rose in the morning in a fasting, non-recumbent position... Blood samples were then obtained via the cubital vein.”: Was there any particular reason for the 2-hour non-recumbent period before blood sampling? A brief justification for this protocol would be useful. If the timing or posture before blood sampling is crucial for measuring RAAS activity, briefly explain its importance.

•The study was approved by the Ethics Committee, however, the approval number is not provided: Please include. State how patient data was anonymized or protected during the study.

•Ensure consistency in terminology. For example, “participants” and “patients” are used interchangeably. Stick to one term for the study population.

•Consider move the subsection “Group and Subgroup” before the inclusion/exclusion criteria. This will give readers a clear understanding of the study's design before reviewing the criteria.

•Consider including a flow diagram that summarizes the patient selection process and subgroup classification.

•Please be consistent in reporting P-values (e.g. rounding to 2 decimal units)

•Method of detection: Please add manufacture and manufacturing country in brackets after “AutoLumo A2000 Plus”

**Results**

“Result” should be changed to “Results”.

Section 3.2 – State clearly what trend (i.e., between which complications or variables) is depicted in the statement “The concentrations of ALD and REN exhibited a consistent trend, while ARR showed an opposite trend to the REN concentration (Fig 1)”

Table 1: Please capitalize “age” and write “Kg/m2” as “kg//m2”; add a space between “Diabetic duration” and “(M)”.

Under “Complication subgroup analysis of the RAAS system”: Please change “cross” to “across”.

**Figure 1**

The analytes are reported in different units of measurement (pg/ml, ng/dL and ng/L). Use a uniform unit of measurement for all analytes to enable easier trend detection.

Caption - “*** P<0.0017”: Suggest changing p-value level to p<0.005.

Include the means ± standard deviation in legend as it is difficult to ascertain this data from the graphs.

**Discussion**

Please change “Discuss” to “Discussion”.

Page 21: Add “A” before “Previous study” and insert a reference for this study.

Page 23 –Consider adding study details, particularly sample size to this sentence “A comprehensive study with a substantial sample size revealed a wide distribution of renin activity in hypertensive patients, particularly among those with diabetes [19].”

We look forward to receiving your revised manuscript.

Kind regards,

Nalini Govender, Ph.D

Academic Editor

PLOS ONE

**Reviewers' comments:**

Reviewer's Responses to Questions

**Comments to the Author**

1. Is the manuscript technically sound, and do the data support the conclusions?

Reviewer #1: Partly

Reviewer #2: Yes

2. Has the statistical analysis been performed appropriately and rigorously? 

Reviewer #1: I Don't Know

Reviewer #2: Yes

3. Have the authors made all data underlying the findings in their manuscript fully available?

Reviewer #1: Yes

Reviewer #2: Yes

4. Is the manuscript presented in an intelligible fashion and written in standard English?

Reviewer #1: No

Reviewer #2: Yes

**5. Review Comments to the Author**

**Reviewer #1:**  In this manuscript, Baohong et al. addressed the link beetwen RAAS activation and glucose metabolism in patients suffering from different T2DM complications.

A large array of observations, in the current literature, shows that dysregulation of RAAS in the metabolic syndrome favors type 2 diabetes. Therefore, the concept of a dual axis of RAAS regarding glucose homeostasis has emerged.

A large body of evidence had already shown that RAAS blockade improves glucose homeostasis and prevents diabetes in patients suffering from the metabolic syndrome, but, currently, drugs targeting RAAS are not given for the purpose of preventing diabetes in patients suffering from metabolic syndrome.

The authors provide a thorough description of RAAS activation both in T2DM patients and in different conditions related to T2DM ( hypetersion, nephropathy, ketoacidosis) which is pretty innovative. Anyhow, in my opinion, in order to make this manuscript more appealing, the authors should include in the analysis the effects of both SGLT2-inhibitors and GLP1 -RA on RAAS. Finally, it would be convenient to mention how ARR interpretation, because of drug interference, could affect clinical practice.

**Reviewer #2:**  Well written and scientifically sound manuscript.

Extensive technical data supports the conclusion as each set of analyzed data is explained in the results section, either tabulated and/or projected with graphical statistics.

Statistical analysis conforms to the typical and most appropriate for the dataset.

The flow of the manuscript is logical and ordered.

MINOR REVISIONS

**Methods Section**

**Study subject**

Discrepancy in total sample size and allocated subgroup

Total sample size should be n=237 (Experimental group and controls). Sample size stated (subheading 2.1) involved a retrospective selection of 191 patients diagnosed with T2DM (Total n=151, but subgroup DMHT=151, DMNT=40 and Controls=46)

Perhaps including the sample number per subgroups (under subheading 2.3) will also assist.

Include ethics number under sub heading 2.2 (end of paragraph)

**Results**

Was a normality test performed initially to determine if sample population was parametric or non-parametric in order to settle with using mean and SD or SEM and interquartile range to represent your data.

In Table 1 and Table 2, p values should not technically be represented as zero (especially when testing a hypothesis). Perhaps represent as p < 0.0001.

p value should be in lowercase and in italics throughout the manuscript.

The normal peripheral blood concentration ranges for normotensive (control) patients:

Angiotensin II = 5 to 35pg/ml

Renin = 0.7 to 3.3ng/ml/hr

Aldosterone = 7 to 30ng/dL

Does your control group conform to these ranges or are you using alternate referenced ranges?

**Discussion**

“Conversely, the absence of such a reaction in DNK indicates impaired juxtaglomerular cell function, leading to no renin release even under dehydration stimulation” – consider rephrasing as your data suggests that renin was detected but concentration levels were decreased.

6. PLOS authors have the option to publish the peer review history of their article (what does this mean? ). If published, this will include your full peer review and any attached files.

**Do you want your identity to be public for this peer review?** For information about this choice, including consent withdrawal, please see our Privacy Policy .

Reviewer #1: **Yes: ** Roberta Poli MD PhD

Reviewer #2: No

---

## [Author Response · Author response to Decision Letter 1]

22 Nov 2024

November 16, 2024

Dear Nalini Govender:

Re: PONE-D-24-34834

Title: Renin-angiotensin-aldosterone system variations in type 2 diabetes mellitus patients with different complications and treatments: implications for glucose metabolism

Thank you for your kind letter of “PLOS ONE PONE-D-24-34834 Revision Request” on October 04, 2024. We thank the editorial requests for the corrections, and we deeply apologized for the considerable inconvenience caused to you due to our errors exist. We have now revised the manuscript according to the concerns raised in the reviews. A detailed copy of the authors' “point-by-point response to the reviewer's comments” is attached, and specific changes addressing reviewers' comments are marked.

C1: General comments: There are several grammatical and punctuation errors throughout the manuscript, which can be revised with the expertise of a language editor.

A1: Great thanks for your comment，and we deeply apologized for our errors exist. We have carefully revised the text, and an English language editor has edited this study. The revised parts were highlighting in the manuscript.

C2: Submission page: Corresponding Author; Order of Authors: The is abbreviated incorrectly. Please correct to abbreviated term “ph.D” to “PhD”; and “CHINA” to “China”.

A2: We apologized for our errors exist, and we have corrected the incorrect abbreviation.

Abstract

C3: Change “Result” to “Results”; “cross” to “across”. “Key words” should be one word, i.e., “Keywords”.

A3: We apologized for our errors exist, and we have corrected the misspellings.

Introduction

C4: Page 1: “The global diabetic population has surged to 537 million and is projected to escalate to 642 million by 2040” - Does this statistic represent both Type 1 and Type 2 diabetes, or just Type 2 diabetes? Clarification on this point would help readers understand the scope of the study.

A4: Great thanks for your comment. This statistic includes both T1DM and T2DM, with T2DM accounting for the vast majority (>90%). We have revised the sentence as “The worldwide population of individuals diagnosed with diabetes mellitus (DM) has surged to 537 million in 2021[1] and is projected to escalate to 642 million by 2040[2].”

C5: Page 2-3: Revise the grammar or reword the sentence “China has witnessed a significant increase in diabetes prevalence in recent years, with the highest number of people with diabetes [3, 4]. Please provide statistics to support this statement. Additionally, is this true for both Type 1 and Type 2 diabetes?

A5: Great thanks for your comment, and we have revised the sentence as “In China, the overall prevalence of diabetes has been on the rise, increasing from 9.7% in 2007 and 2010 to 10.4% in 2013, and further to 11.2% in 2017[3-7]. China ranks number one with an estimate of 129.8 million adults with diabetes by 2017[7]; among them, type 2 diabetes mellitus (T2DM) accounts for the vast majority (>90%) [8].”

C6: The statement “Research indicates that the activation status of the renin-angiotensin aldosterone system (RAAS) undergoes alterations in diabetic patients with various complications [9-11]” Please specify if this is for Type 1 or Type 2 diabetic patients or inclusive of both types.

A6: Great thanks for your comment, and we have revised the sentence as “Research indicates that the activation status of the renin-angiotensin-aldosterone system (RAAS) undergoes alterations in Type 1 and Type 2 diabetes mellitus (T1DM/T2DM) patients with various complications [12-14].”

C7: Please change “rate” to “rates”. Please insert “levels” after “elevated HbA1c”. Please change “blood glucose control deteriorated” to “blood glucose levels deteriorated”

A7: We apologized for our errors exist, and we have corrected the misspellings and improper expressions.

C8: The statement “A comprehensive study with a substantial sample size revealed a wide distribution of renin activity in hypertensive patients, particularly among those with diabetes [19].” Should be expanded to include sample size and more data to enhance the relevance

A8: Great thanks for your comment, and we have revised the statement as“Two systematic studies of plasma renin activity, conducted on a substantial community sample of hypertensive patients (n=1660[22] and n=4170[23], respectively), revealed a wide distribution of activity level, particularly among diabetic patients.”

C9: Please add reference to support this statement “Accurate measurement of renin levels not only helps evaluate patient compliance with prescribed therapies but also assists clinicians in optimizing treatment strategies.”

A9: Great thanks for your comment, and we have added references 19 and 24 to support our statement.

Method

C10: Study subjects: Be consistent and use two decimal places for “average age of 52.3 years”

A10: Great thanks for your comment. We have modified the average age to 52.30 years.

C11: Verbal consent is indicated, however written informed consent is most applicable, especially when dealing with personal and sensitive health data. Was verbal consent the only form of consent? If so, please clarify why this was deemed appropriate. Also, consider adding whether the study followed any privacy guidelines, such as anonymizing data.

A11: The study employed a retrospective design, involving the selection of 237 participants from the First People's Hospital of Pingdingshan between April 2023 and May 2024. The data were accessed for research purposes on May 6, 2024. All participants had previously provided written informed consent during their examination; however, due to the retrospective nature of the study, some participants had been discharged from the hospital by the time of data collection. Consequently, we contacted these individuals via telephone to obtain verbal informed consent once more. During or after the data collection, personal identifiers such as names were not utilized; instead, a numerical identifier was employed to anonymize the information. The data collected was intended solely for academic research purposes, and all personal information was treated with the utmost confidentiality. Owing to article length limitations, we apologized for the vague expression in the paper, so the process of informed consent and anonymizing data to protect personal privacy was added in 2.1 and 2.3 section of the Method.

C12: Why are individuals undergoing treatment with aldosterone receptor antagonists and direct renin inhibitors excluded? Are these treatments expected to interfere with RAAS measurements? A brief justification would clarify this decision so that other readers understand.

A12: Great thanks for your comment, and we have added the following justification: “Aldosterone receptor antagonists and direct renin inhibitors significantly affect aldosterone and renin levels, subsequently influencing the aldosterone-to-renin ratio [26]”.

C13: The control group consists of “46 healthy individuals.” How was the absence of diabetes or hypertension verified? Were any potential confounders (such as family history of diabetes or hypertension) accounted for when selecting the control group?

A13: Great thanks for your comment, and we apologized for the vague expression in the paper. We have added the process of control group selection that was shown in Fig1 and revised the following statement: “A cohort of 46 healthy individuals was selected as the control group, all of whom attended our hospital for routine physical examinations during the same period. These control individuals had never suffered from diabetes or hypertension and had no family history of these conditions” in 2.1 section of the Method.

C14: Regarding the statement “The individuals rose in the morning in a fasting, non-recumbent position... Blood samples were then obtained via the cubital vein.”: Was there any particular reason for the 2-hour non-recumbent period before blood sampling? A brief justification for this protocol would be useful. If the timing or posture before blood sampling is crucial for measuring RAAS activity, briefly explain its importance.

A14: Great thanks for your comment, and we added the following justification: “The patient's posture before blood sampling is crucial for measuring RAAS activity, and the aldosterone-to-renin ratio (ARR) test is most sensitive when samples are collected in the morning after patients have been up (sitting, standing, or walking) for at least 2 hours and seated for 10 minutes [26]. Consequently, all patients in this study adhered to this criterion for blood collection.” in 2.4 section of the Method.

C15: The study was approved by the Ethics Committee; however, the approval number is not provided: Please include. State how patient data was anonymized or protected during the study.

A15: Great thanks for your comment. The ethical approval number has been provided, and the process of anonymizing data to protect personal privacy was added in 2.1 and 2.3 section of the Method.

C16: Ensure consistency in terminology. For example, “participants” and “patients” are used interchangeably. Stick to one term for the study population.

A16: We have adhered to the terminology "patients" to refer to the study population.

C17: Consider move the subsection “Group and Subgroup” before the inclusion/exclusion criteria. This will give readers a clear understanding of the study's design before reviewing the criteria.

A17: Great thanks for your comment. We have moved the subsection “Group and Subgroup” before the inclusion/exclusion criteria.

C18: Consider including a flow diagram that summarizes the patient selection process and subgroup classification.

A18: Great thanks for your comment. We have included a flow diagram that illustrates patient selection and subgroup classification, as shown in Fig 1.

C19: Please be consistent in reporting P-values (e.g. rounding to 2 decimal units)

A19: We have maintained consistency in reporting P-values to three decimal places.

C20: Method of detection: Please add manufacture and manufacturing country in brackets after “AutoLumo A2000 Plus”

A20: The manufacture and manufacturing country (Autobio, China) was added in brackets after “AutoLumo A2000 Plus”.

Results

C21: “Result” should be changed to “Results”.

A21: We apologized for our errors exist, and we have corrected the misspellings.

C22: Section 3.2 – State clearly what trend (i.e., between which complications or variables) is depicted in the statement “The concentrations of ALD and REN exhibited a consistent trend, while ARR showed an opposite trend to the REN concentration (Fig1)”

A22: Great thanks for your comment. We have revised the statement as “Among the DN, DK, DNK, OCHT, NCHT, OCNT, and NCNT subgroups, the concentrations of ALD and REN exhibited a consistent trend, while ARR showed an opposite trend to the REN concentration (Fig 2)”.

C23: Table 1: Please capitalize “age” and write “Kg/m2” as “kg/m2”; add a space between “Diabetic duration” and “(M)”.

A23: We apologized for our errors exist, and we have corrected these errors.

C24: Under “Complication subgroup analysis of the RAAS system”: Please change “cross” to “across”.

A24: We apologized for our errors exist, and we have corrected the misspellings.

Figure 1

C25: The analytes are reported in different units of measurement (pg/ml, ng/dL and ng/L). Use a uniform unit of measurement for all analytes to enable easier trend detection.

Caption - “*** P<0.0017”: Suggest changing p-value level to p<0.005.

Include the means ± standard deviation in legend as it is difficult to ascertain this data from the graphs.

A25: Great thanks for your comment. ng/dL=10pg/mL=10ng/L. We used the uniform unit(ng/L) to measure AII, ALD, REN. Consequently, aldosterone levels were increased tenfold, leading to a revised calculation formula for ARR, expressed as ARR = ALD / (REN × 10) (2.4 section of the Method). All data presented in this article have been reanalyzed using SPSS and revised accordingly.

We have changed “***p<0.0017” to “*** p<0.001” in the caption.

Additionally, we have included the medians of AII, ALD, REN, and ARR in the legend, as shown in Fig 2.

Discussion

C26: Please change “Discuss” to “Discussion”.

A26: We apologized for our errors exist, and we have corrected the misspellings.

C27: Page 21: Add “A” before “Previous study” and insert a reference for this study.

A27: We apologized for the vague expression in the paper, and we have revised “Previous study” as “Our study”.

C28: Page 23 –Consider adding study details, particularly sample size to this sentence “A comprehensive study with a substantial sample size revealed a wide distribution of renin activity in hypertensive patients, particularly among those with diabetes [19].”

A28: Great thanks for your comment, and we have revised the statement as “Two systematic studies of plasma renin activity, conducted on a substantial community sample of hypertensive patients (n=1660[22] and n=4170[23], respectively), revealed a wide distribution of activity level, particularly among diabetic patients.”

Again, we would like to thank you for taking the time to review our manuscript. We would be most grateful for your kind consideration of this revised manuscript for potential publication in PLOS ONE.

We look forward to hearing from you at your earliest convenience.

Sincerely yours,

Ningning Wang

Corresponding author:

Baohong Yue, Professor, PhD

Address: Department of Laboratory Medicine, The First Affiliated Hospital of Zhengzhou University, Zhengzhou, P. R. China

Responses to the Reviewer’s comments:

Re: PONE-D-24-34834

Title: Renin-angiotensin-aldosterone system variations in type 2 diabetes mellitus patients with different complications and treatments: implications for glucose metabolism

We would like to acknowledge the reviewer’s insightful comments and valuable suggestions. In addition, we thank the reviewer for helping us improve the present manuscript. We have addressed the concerns raised by the reviewer as detailed below.

Reviewer #1

C1: In this manuscript, Baohong et al. addressed the link between RAAS activation and glucose metabolism in patients suffering from different T2DM complications.

A large array of observations, in the current literature, shows that dysregulation of RAAS in the metabolic syndrome favors type 2 diabetes. Therefore, the concept of a dual axis of RAAS regarding glucose homeostasis has emerged.

A large body of evidence had already shown that RAAS blockade improves glucose homeostasis and prevents diabetes in patients suffering from the metabolic syndrome, but, currently, drugs targeting RAAS are not given for the purpose of preventing diabetes in patients suffering from metabolic syndrome.

The authors provide a thorough description of RAAS activation both in T2DM patients and in different conditions related to T2DM (hypertension, nephropathy, ketoacidosis) which is pretty innovative. Anyhow, in my opinion, in order to make this manuscript more appealing, the authors should include in the analysis the effects of both SGLT2-inhibitors and GLP1 -RA on RAAS. Finally, it would be convenient to mention how ARR interpretation, because of drug interference, could affect clinical practice.

A1: We are appreciated to note the favorable comments of reviewer in the comments.

Great thanks for your comment. We have added “Antidiabetic treatment of T2DM patients and its influence on RAAS” in Results section and S3, S4 Table. All T2DM

patients received antidiabetic treatment, among which 95 (49.7%) took SGLT-2is, 40 (20.9%) took GLP-1RAs injection and 55(28.8%) took DPP-4is. The subsequent analysis revealed that SGLT-2is, GLP-1RAs, DPP-4is and other glucose-lowering agents had no statistically significant effect on the RAAS system (p>0.05). We also add this point of view in Abstract and Discussion section.

We incorporated glucose-lowering treatment into the covariate model for the multiple linear regression analysis of RAAS and glucose metabolism. The results indicated that the correlation between RAAS and glucose metabolism was not influenced by any antidiabetic treatments (Table 3). We also add this point of view in Abstract and Discussion section.

Reviewer #2

C1: Well written and scientifically sound manuscript.

Extensive technical data supports

---

## [Editor Report · Decision Letter 1]

5 Dec 2024

Renin-angiotensin-aldosterone system variations in type 2 diabetes mellitus patients with different complications and treatments: implications for glucose metabolism

PONE-D-24-34834R1

Dear Dr. Yue,

We’re pleased to inform you that your manuscript has been judged scientifically suitable for publication and will be formally accepted for publication once it meets all outstanding technical requirements.

Kind regards,

Nalini Govender, Ph.D

Academic Editor

PLOS ONE
---

## [Editor Report · Acceptance letter]

PONE-D-24-34834R1

PLOS ONE

Dear Dr. Yue,

I'm pleased to inform you that your manuscript has been deemed suitable for publication in PLOS ONE. Congratulations! Your manuscript is now being handed over to our production team.

Kind regards,

on behalf of

Prof Nalini Govender

Academic Editor

PLOS ONE